☼ PLOS | ONE

# Jmjd1c is dispensable for healthy adult hematopoiesis and Jak2$^{V617F}$-driven myeloproliferative disease initiation in mice

**Hans F. Staehle, Johannes Heinemann, Albert Gruender, Anne M. Omlor, Heike Luise Pahl$^☉$, Jonas Samuel Jutzi**[iD]$^{☉¤}$*

Division of Molecular Hematology, University Medical Center Freiburg, Faculty of Medicine, University of Freiburg, Freiburg, Baden-Württemberg, Germany

☉ These authors contributed equally to this work.
¤ Current address: Brigham and Women's Hospital, Harvard Institute of Medicine, Boston, Massachussetts, United States of America
* jjutzi@bwh.harvard.edu

**Data Availability Statement:** All relevant data are within the manuscript and its Supporting Information files.

## Abstract

The histone demethylase JMJD1C is overexpressed in patients with myeloproliferative neoplasms (MPNs) and has been implicated in leukemic stem cell function of MLL-AF9 and HOXA9-driven leukemia. In the emerging field of histone demethylase inhibitors, JMJD1C therefore became a potential target. Depletion of Jmjd1c expression significantly reduced cytokine-independent growth in an MPN cell line, indicating a role for JMJD1C in MPN disease maintenance. Here, we investigated a potential role for the demethylase in MPN disease initiation. We introduced a Cre-inducible JAK2$^{V617F}$ mutation into Jmjd1c knockout mice. We show that Jmjd1c is dispensable, both for healthy hematopoiesis as well as for JAK2$^{V617F}$-driven MPN disease initiation. Jmjd1c knockout mice did not show any significant changes in peripheral blood composition. Likewise, introduction of JAK2$^{V617F}$ into Jmjd1c$^{-/-}$ mice led to a similar MPN phenotype as JAK2$^{V617F}$ in a Jmjd1c wt background. This indicates that there is a difference between the role of JMJD1C in leukemic stem cells and in MPN. In the latter, JMJC domain-containing family members may serve redundant roles, compensating for the loss of individual proteins.

## Introduction

The pathophysiology of myeloproliferative neoplasia (MPN) remains incompletely understood, despite the discovery of disease defining mutations such as JAK2$^{V617F}$ and C-terminal alterations of CALR. In particular, it is not clear which effectors or pathways are required for disease initiation or maintenance in addition to aberrant JAK/STAT signaling. Elucidating additional oncogenic determinants is clinically meaningful in light of the limited therapeutic efficacy of JAK2 inhibition in these disorders. A rational search for drug combinations can be informed and accelerated by pre-clinical investigation of potential targets.

We have previously shown that the histone 3 mono and dimethyl-specific demethylase JMJD1C is overexpressed in MPN patients [1]. JMJD1C participates in an auto-regulatory loop, as it is both a target of the transcription factor NFE2, overexpressed in the large majority

**Funding:** This work was supported by grants from the Deutsche Forschungsgemeinschaft (Pa 611/9-1 and SFB 992, project B02, to H.L. Pahl and Ju 3104/1-1 to J.S. Jutzi). J.S. Jutzi was funded by the Excellence Initiative of the German Federal and States Governments (GSC-4, Spemann Graduate School). H.F.S. was supported by the M.D. mentoring program MOTI-VATE funded by the Else Kröner-Fresenius-Stiftung (EKFS).

**Competing interests:** The authors have declared that no competing interests exist.

of MPN patients, and also binds the NFE2 promoter, thereby enhancing NFE2 expression. Moreover, depletion of JMJD1C in JAK2$^{V617F}$-expressing BAF/3 cells significantly reduced cell proliferation and this effect was more pronounced during cytokine-independent growth. These data suggest that JAK2$^{V617F}$-mediated proliferation and growth-factor independence is at least partially dependent on the presence of JMJD1C.

MPN patients are at risk of transformation to acute leukemia, a life-threatening disease exacerbation that is often refractory to treatment. JMJD1C has recently been shown to play a critical role in the survival of acute myeloid leukemia (AML) cells. Depletion of JMJD1C severely impaired proliferation of ten different AML cell lines, carrying various oncogenic fusion genes or mutations, including AML-ETO, PML-RAR, FLT3-ITD, t(3;3) with Evi-1 over-expression and JAK2$^{V617}$ among others [2]. Moreover, JMJD1C is required for leukemic stem cell self-renewal in murine models of both MLL-AF9 and HOX-A9-driven AML [3]. In these mice, a genetic knockout of JMJD1C decreased the frequency of leukemic stem cells and caused differentiation. This observation is especially intriguing in light of the finding that in this model, hematopoietic stem cells (HSCs) appeared less effected than leukemic stem cells (LSCs).

We therefore tested the hypothesis that JMJD1C is required for JAK2$^{V617F}$-driven MPN disease initiation by engineering mice conditionally expressing the active kinase in absence or presence of the histone demethylase.

## Materials and methods

### Generation of *Jmjd1c* knockout mice

Knockout first mice (*Jmjd1c$^{tm1a(EUCOMM)Wtsi}$*) were purchased from EUCOMM (ID-No. 71834). These animals referred to as "*Jmjd1c-k*" mice, contain the tm1a allele which introduces a lacZ exon-trap (Fig 1A). A gene trap cassette, following exon 8 of the *Jmjd1c* locus, contains an additional splice acceptor (SA), which should link the spliceosome to an artificial polyade-nylation sequence (pA). Translation of *lacZ* as an independent polypeptide occurs via an internal ribosomal entry site (IRES). Furthermore, the tm1a allele contains a selection cassette allowing the expression of a neomycin resistance gene (neo) under the control of the human ß-actin promoter (Bact). Two Frt sites flank the two cassettes and allow their removal by the application of the FLP recombinase. Similarly, loxP sites surround exons 9 and 10 and represent recognition sequences for the Cre recombinase. *Jmjd1c-k* mice were bred with *FLPe* expressing mice (*129S4/SvJaeSor-Gt(ROSA)26Sor$^{tm1(FLP1)Dym}$/J*) to remove the lacZ and neo markers, yielding mice with a floxed *Jmjd1c* locus, termed "*Jmjd1c-p*". Mating with *Cre deleter* mice (*B6.C-Tg(CMV-cre)1Cgn/J*) generated animals deleted for exons 9 and 10 of *Jmjd1c*, ("*Jmjd1c-d*"). Both, *FLPe* and *Cre deleter* mice, were kind gifts of Prof. R. Schüle, University Medical Center Freiburg.

### Generation of *Jmjd1c* knockout mice carrying the *Jak2$^{V617F}$* allele

Conditional floxed *Jak2$^{+/L2}$* knock-in (ki) mice (*Jak2$^{tm2.2Jlvl}$*) have been previously described [4] and were a generous gift of Jean-Luc Villeval. They are heterozygous for a construct that allows expression of *Jak2$^{V617F}$* in presence of the *Cre recombinase*. *Jmjd1c$^{d/d}$* mice were crossed with both *Jak2$^{+/L2}$* and *Mx1-Cre$^{+/Cre}$* transgenic mice (*B6.Cg-Tg(Mx1-Cre)1Cgn/J*, JAX stock No. 003556) to generate animals carrying an inducible *Jak2$^{V617F}$* mutation in the context of either a wt *Jmjd1c* allele (*Jak2$^{+/L}$ Mx1-Cre$^{+/Cre}$ Jmjd1c$^{+/+}$*, called *Jak2$^{V617F}$*) or of a *Jmjd1c* knockout (*Jak2$^{+/L}$ Mx1-Cre$^{+/Cre}$ Jmjd1c$^{d/d}$*, called *Jak2$^{V617F}$ Jmjd1d$^{d/d}$*). Hematological characterization was performed on cohorts of *Jak2$^{V617F}$* and *Jak2$^{V617F}$ Jmjd1d$^{d/d}$* mice, induced at 7 weeks of age by i.p. pI:pC injections (Sigma-Aldrich, No. P1530) administered 3 times within a period of 7 days.

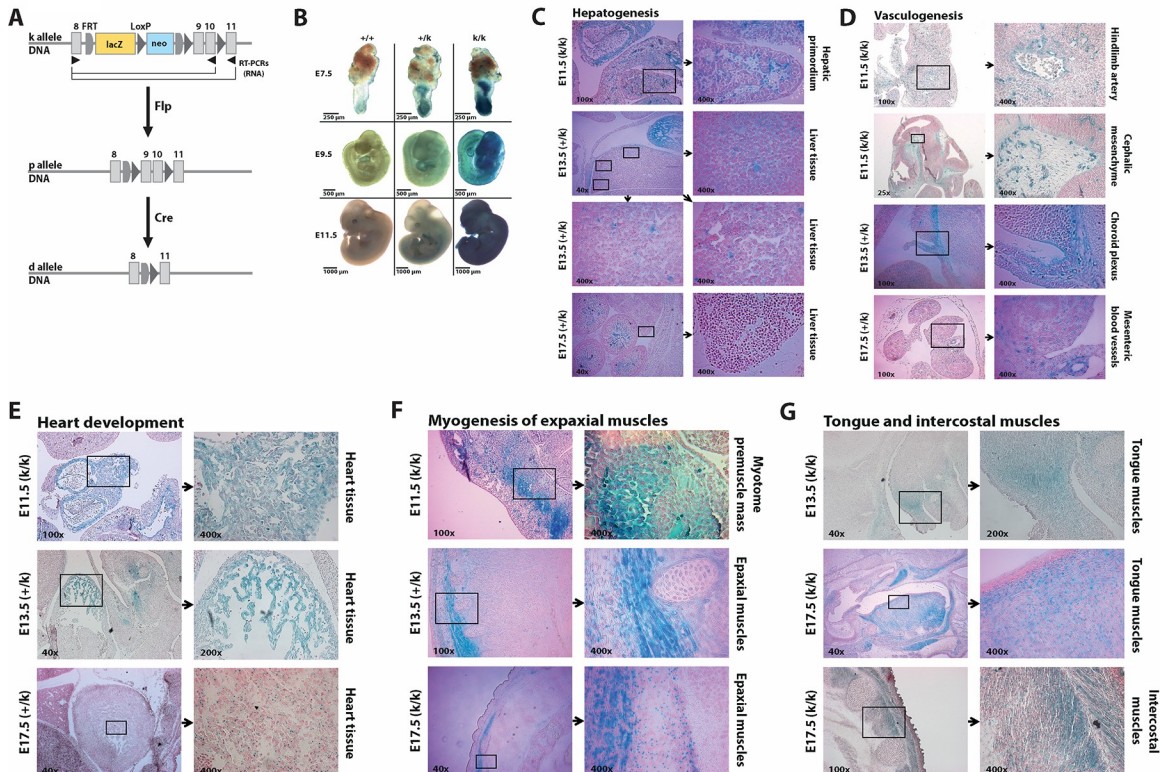

**Fig 1. Expression sites of *Jmjd1c* during embryonic development.** (**A**) Schematics of the *Jmjd1c* allele status. Knockout first mice (*Jmjd1c*-k allele) were crossed with *FLPe* mice to create conditional knockout mice (*Jmjd1c*-p allele). Further matings with *Cre deleter* mice were performed to excise exons 9 and 10 (*Jmjd1c*-d allele). Numeration of exons is shown in S1 Fig. (**B**) Beta-galactosidase staining of whole-mount mouse embryos at day E7.5, E9.5 and E11.5 of embryonic development. Scale bars: E7.5 = 250 μm, E9.5 = 500 μm, E11.5 = 1000 μm. (**C-G**) Sections showing beta-galactosidase signals during hepatogenesis (**C**), vasculogenesis (**D**), heart development (**E**) and myogenesis of skeletal muscles (**F+G**). E11.5 and E13.5 whole mouse embryos were beta-galactosidase stained with subsequent formalin-fixation, paraffin embedding and sectioning, while E17.5 were first frozen followed by beta-galactosidase staining of the cryosections. All sections were counterstained with Nuclear Fast Red.

## Animal housing and protection

All experiments conducted on mice were approved by the Environment and Consumer Protection Agency of the State of Baden-Württemberg, Germany (G-17/59). The reviewing animal ethics committee consisted of lay people and animal welfare experts (veterinarians). Mice were maintained under specific pathogen-free conditions at the research mouse facility of the University Medical Center Freiburg. Lighting was adjusted to the circadian rhythm of the animals and temperature was kept between 20 and 23°C. Mice lived in Type2Long cages, enriched by nesting material such as litter, tunnels and paper towels. Mice had permanent access to water and food (KLIBA NAFAG, Switzerland), which was changed every week or earlier if necessary. Animal health and behavior was monitored once daily by care takers and 5 days per week by research personnel. A special training in animal care and handling (FELASA B certificate) was mandatory for all staff working with mice.

The results of this study are based on 62 mice. Prior to the experiments, humane endpoints were determined to avoid pain and distress of the animals. These include local infections, decrease in body weight, large tumors, bleeding, decrease of activity, paralysis, etc. Once animals reached endpoint criteria, they were sacrificed on the same day. The phenotype of *Jmjd1c^{d/d}* mice was investigated for 40 weeks. Jmjd1c^{k/k} and *Jak2^{V617F} Jmjd1d^{d/d}* mice were

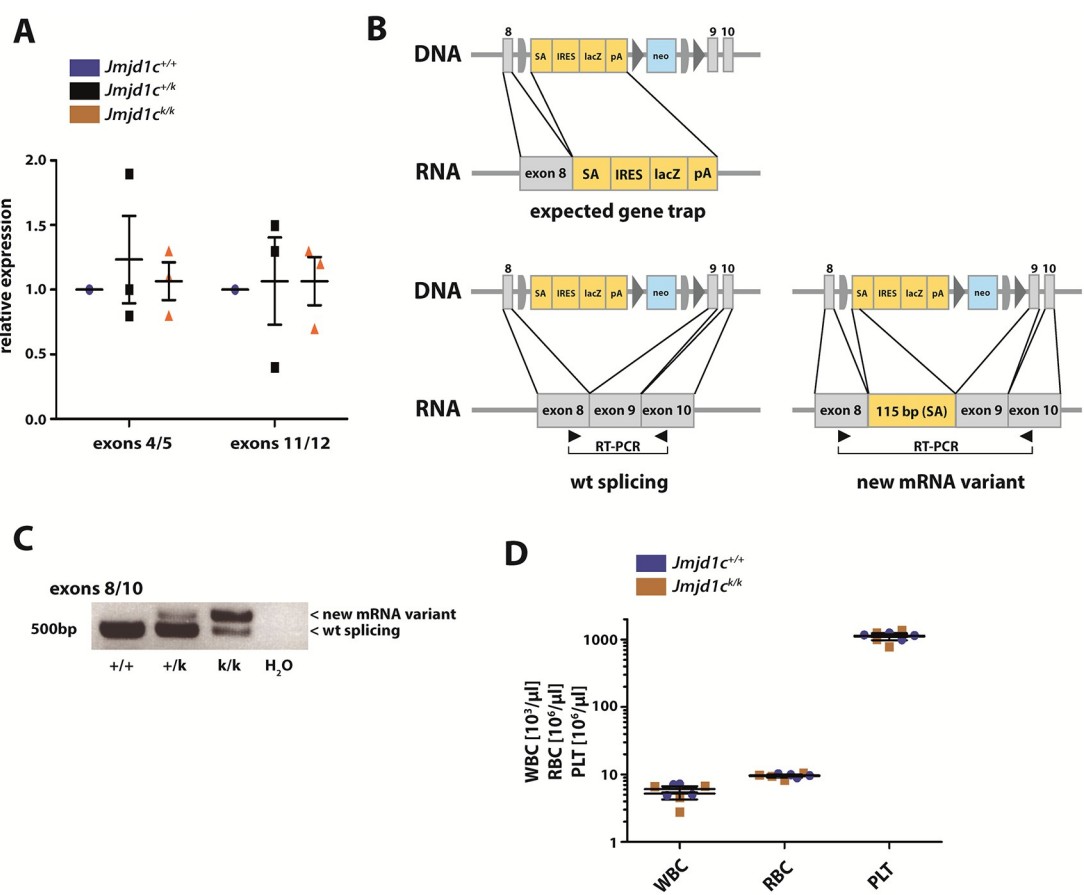

**Fig 2. The knockout first approach leads to alternative splicing and a new mRNA variant in *Jmjd1c^{k/k}* mice. (A)** RT-qPCRs spanning exons 4/5 and 11/12 with cDNA obtained from whole embryos. **(B)** Expected and observed mRNA variants in mice carrying the k allele. **(C)** RT-PCR spanning exons 8/10 with cDNA obtained from whole embryos. Bands were cut out and extracted DNA was sequenced (S5 Fig). **(D)** PB count of 12-week-old *Jmjd1c^{+/+}* and *Jmjd1c^{k/k}* mice (n = 4 per genotype). Mann-Whitney U test was used for statistical calculations. Data are represented as mean +/- SEM.

investigated for 12 weeks. Only one wild type control mouse was found dead without prior symptoms and the cause of death could not be determined. All other animals were sacrificed by carbon dioxide euthanasia followed by cervical dislocation.

### Genotyping-PCR, RT-PCR and RT-qPCR

Ear punches were used to extract gDNA for PCR genotyping (S2 and S3 Figs; GeneJET Genomic DNA Purification Kit (Thermo Scientific, No. K0722)). From murine peripheral blood (PB), BM or testicular tissue both gDNA and RNA were extracted (AllPrep DNA/RNA Micro Kit (Qiagen, No. 80204)). Reverse transcription was performed using 400 ng RNA in the Taq-Man Reverse Transcription kit (Applied Biosystems, No. 4368813) and the resulting cDNA assayed for *Jmjd1c* exon usage by PCR (Figs 2A+2C and 3). Primer sequences are shown in S1 Table. Original, minimally cropped and adjusted gel images are shown in S4 Fig.

RT-qPCR was used to determine the efficiency of gene trapping in *Jmjd1c^{k/k}* mice. The gene trap should cause splicing of exon 8 to an artificial splice acceptor (SA), which is linked to a poly-adenylation site (pA). *Jmjd1c* mRNA in *Jmjd1c^{k/k}* mice should therefore lack all exons downstream of exon 8. Quantifying the expression of exons 4/5 and exons 11/12, thus

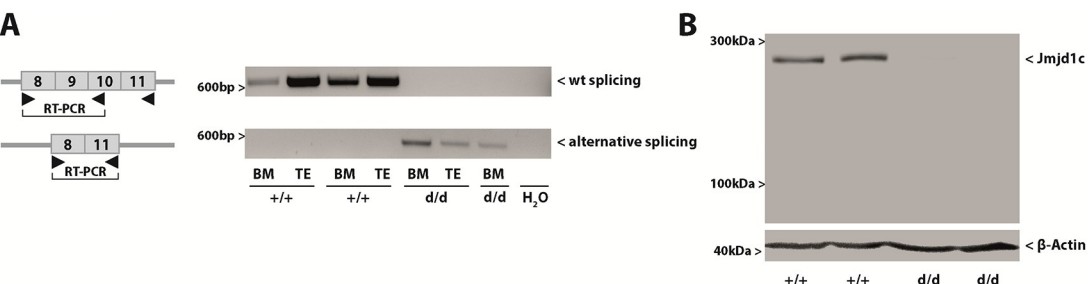

**Fig 3. Creation and validation of *Jmjd1c$^{d/d}$* mice.** **(A)** RT-PCRs using material obtained from bone marrow (BM) and testicular tissue (TE). **(B)** Western blotting showing the expression of *Jmdj1c* in *Jmjd1c$^{+/+}$* and *Jmjd1c$^{d/d}$* mice. Proteins were extracted from testicular tissue.

determines the efficiency of gene trapping (Fig 2A). Commercially available reagents were used for exons 4/5 (Applied Biosystems, Mm01150348_g1), exons 11/12 (Applied Biosystems, Mm01150330_m1) and the housekeeping gene murine β-2-microglobulin (Applied Biosystems, Assay on Demand). RT-qPCRs were performed in duplicates using a LightCycler 480 (Roche). Data were analyzed with the -ΔΔCT method.

## Western blotting and antibodies

Murine bone marrow or testicular tissue was homogenized with a 100 μm cell strainer and isolated cells were resuspended in lysis buffer (20 mM Tris-HCl, pH 7.5, 150 mM NaCl, 5 mM EDTA, pH 8.0 and 1% Triton X-100 in ddH$_2$O) followed by vigorous vortexing for 10 min at 4°C. Cell debris was removed by centrifugation. Before proceeding with western blotting, protein concentrations were determined by Bradford assay (Bio-Rad, Hercules, CA, USA, No. 500–0007). Blots were probed with a primary antibody (polyclonal, rabbit) directed against Jmjd1c (Merck Millipore, No. 17–1026), subsequently stripped and re-probed for β-Actin (Sigma-Aldrich, No. A5441) to control for equal loading. Immunocomplexes were detected by enhanced chemiluminescence using an Intas Imager for visualization. Original, minimally cropped and adjusted blot images are shown in S4 Fig.

## Blood draws

Blood samples from mice were taken via puncture of retrobulbar veins. A heparin-coated 10 μl capillary was pushed forward into the inner lid angle behind the ocular bulb. With soft rotational movements, some of the small blood vessels in the orbital cavity were disrupted. Depending on the weight of the mouse, 100–150 μl blood was collected in heparin-coated 300 μl Microvette tubes. After removing the capillary, an adequate pressure on the eye was used to stop continued bleeding. Suffering and distress of the animals were minimized by isoflurane narcosis and a heating mat was used to avoid hypothermia. The whole procedure including the narcosis took approximately 2–3 minutes per animal. Analysis of the complete blood count (CBC) was performed on an Advia 120 system (Siemens) and an Animal Blood Counter (Scil Vet).

## FACS analysis

Flow cytometry experiments were performed using a BD FACS Fortessa. Populations of mature blood cells were identified by staining PB and BM for B220 (BioLegend, clone RA3-6B2), CD3 (Thermo Scientific, clone 145-2C11), Gr1 (BioLegend, clone RB6-8C5) and Mac1

(BioLegend, clone M1/70). Analysis of erythroid precursors in PB and BM was conducted with antibodies against Ter-119 (BioLegend, clone TER-119) and CD71 (BioLegend, clone R17217). Stem and progenitor cells in BM were identified as previously described [5–7] by staining with a cocktail against lineage markers (BioLegend, B220, CD3, Gr1, Mac1 and Ter119) as well as for c-Kit (eBioscience, clone 2B8), Sca1 (BioLegend, clone D7), CD34 (Bio-Legend, clone MEC14.7), Fc-γ-II/III-R (eBioscience, clone 93), Thy1.1 (BioLegend, clone OX7) and Flt3 (eBioscience, clone A2F10). Gating strategies were determined by fluorescence minus one staining [8].

## β-Galactosidase staining

Embryos day E13.5 and younger were stained as a whole mount. Briefly, embryos were washed in x-gal buffer (5mM EGTA, 2mM MgCl$_2$, 0.01% Natrium Deoxycholat, 0.02% NP40 in DPBS with Ca$^{2+}$/Mg$^{2+}$) and fixed in solution Ia (1% formaldehyde and 0.2% glutaraldehyde in x-gal buffer) for 30–60 minutes at 4˚C. Subsequently, embryos were stained in solution Ib (5 mM K$_3$Fe(Cn)$_6$, 5 mM K$_4$Fe(Cn)$_6$, 1 mg/ml X-gal in x-gal buffer) overnight at 37˚C in the dark followed by fixation in 4% formalin and embedding in paraffin as previously described [8]. Sections were counterstained with Nuclear Fast Red (Sigma-Aldrich, No. N3020).

E17.5 embryos were processed by cryopreservation and cryosectioning. Whole embryos were prepared for cutting by fixation in 4% paraformaldehyde (PBS based) for 30 minutes on ice. Following washes in ice-cold PBS, the samples were incubated in 30% sucrose (PBS based) for 4 hours at 4˚C, transferred to a 1:1 mix of OCT (optimal cutting temperature compound, Sakura Finetek, No. S0378) and 30% sucrose (in PBS) and incubated overnight at 4˚C. Subsequently, embryos were embedded in OCT and frozen on dry ice. 8–10 μm sections were cut, fixed in solution IIa (2% paraformaldehyde and 0.2% glutaraldehyde in washing buffer) for 5 minutes at 4˚C, washed (2mM MgCl$_2$ and 0.02% NP40 in PBS at RT) and stained in solution IIb (10 mM K$_3$Fe(Cn)$_6$, 10 mM K$_4$Fe(Cn)$_6$, 1 mg/ml X-gal in washing buffer) overnight at 37˚C in the dark. Slides were counter stained with Nuclear Fast Red (Sigma-Aldrich, No. N3020).

## Hematoxylin and eosin (H&E) staining

For histopathological analysis, murine femur and spleen samples were fixed in 4% formalin overnight at room temperature. Decalcification of femora was performed in 10% buffered eth-ylene-diamine tetra-acetic acid (EDTA), pH 7.2. Organs were embedded in paraffin as previously described [8] and sections were stained with H&E.

## Statistical analysis

GraphPad PRISM version 6 was used to carry out the statistical analysis. Data of two groups were analyzed using the Mann-Whitney U test. Survival analysis was conducted with the Log-rank (Mantel-Cox) test. Units, properties of distribution, p-values and n-numbers are specified in the figure legends. In a few cases, data points were lost due to technical issues or due to the premature death of mice (S2 Table).

## Results

The emergence of histone demethylases as novel targets in the treatment of myeloid malignan-cies is highlighted by the fact that several LSD1 inhibitors are already under clinical investiga-tion [9] and JMJC domain-containing demethylase inhibiting substances are in development. While our previous results indicate a possible role for *JMJD1C* in the pathophysiology of MPN

(1), its function in these disorders has not been investigated. Moreover, its role in healthy hematopoiesis remains incompletely understood and developmental expression of *JMJD1C* has not been examined.

We therefore depicted *JMJD1C* expression at five stages of embryogenesis, between day E7.5 and E17.5. From the EUCOMM consortium, we obtained *Jmjd1c* "knockout-first mice" (Fig 1A, top, k allele). This construct is predicted to result in a complete *Jmjd1c* knockout (ko), as a novel splice site is generated, which causes splicing of exon 8 to the inserted cassette that contains a splice acceptor site and a beta galactosidase (lacZ) gene followed by a poly-adenylation site (pA). In *Jmjd1c^{+/k}* and *Jmjd1c^{k/k}* mice, beta-galactosidase staining therefore accurately reflects transcription off the *Jmjd1c* promoter, visualizing sites of Jmjd1c expression (Fig 1B).

On day E11.5, β-galactosidase signals are present in the urogenital ridge close to the dorsal aorta and the subcardinal vein (S6 Fig) as well as in the hepatic primordium (Fig 1C). All three represent sites of hematopoiesis during this stage of embryogenesis. Moreover, we detected strong β-galactosidase signals in the vascular system throughout day E11.5 embryos, for instance, around the hindlimb artery, in the cephalic mesenchyme, in the choroid plexus and in the mesentery (Fig 1D). These data suggest that Jmjd1c plays a previously unrecognized role in vasculogenesis.

The strongest *Jmjd1c* expression was found in myogenous tissues, for instance in the developing heart of E11.5, E13.5 and E17.5 embryos (Fig 1E). Moreover, in day E11.5 embryos β-galactosidase signals indicating *Jmjd1c* expression are present in the myotome premuscle mass, which forms the epaxial muscles among others (Fig 1F). We also detected strong *Jmjd1c* signals in the tongue and in the intercostal muscles of E13.5 and E17.5 embryos (Fig 1G and Table 1). Expression of *JMJD1C* has been shown to repress neural differentiation in human embryonic stem cells (ESCs) [8]. Accordingly, β-galactosidase signals were absent in neuronal tissues, for example in the developing cerebrum, as well as in lung and the developing intestinal tract (S6 Fig).

Because we observed *Jmjd1c* expression at embryonic sites of hematogenesis, we investigated the effect of *Jmjd1c* deletion on steady-state, adult hematopoiesis. Animals homozygous for the k-allele (*Jmjd1c^{k/k}*, Fig 1A), which introduces the novel splice acceptor following exon 8, should retain expression of the proximal 8 exons but no longer express exons 9 and beyond which contain the catalytic part of the protein. However, two different quantitative real-time

**Table 1. Summary of the expression analysis of *Jmjd1c* during embryogenesis.**

| Organ | E11.5 | E13.5 | E17.5 |
|---|---|---|---|
| Heart development | ++ | ++ | + |
| Hematopoiesis | + | + | - |
| Hepatogenesis | - | - | - |
| Intestinal tract | - | - | ++ [a] |
| Kidney development | - | - | - |
| Neurogenesis | - | - | - |
| Respiratory system | - | - | - |
| Skeletal muscles | ++ | ++ | ++ |
| Vasculogenesis | ++ | ++ | + |

Sections of beta-galactosidase stained embryos were evaluated for beta-galactosidase intensity in the different tissues during embryonic development. Representative sections are illustrated in Fig 1 and S6 Fig. No expression (-), weak expression (+), strong expression (++).

[a] Staining signal in the intestine at time point E17.5 is caused by background beta-galactosidase activity (S6E Fig).

PCRs showed that mRNA-expression of the proximal exons 4 and 5 and the distal *Jmjd1c* exons 11 and 12 is equal between *Jmjd1c^{+/+}* and *Jmjd1c^{k/k}* animals (Fig 2A). Further RT-PCR experiments revealed two observations. Firstly, the gene trap did not function efficiently as a substantial amount of normally spliced, wild-type mRNA remained detectable in *Jmjd1c^{k/k}* animals (Fig 2B+2C). This was due to inefficiency of the splice acceptor used in the "knockout-first" construct. Secondly, a new mRNA variant was detected that contained 115 bp of the "knockout-first" cassette, yielding a larger mRNA variant (Fig 2B+2C). In addition to the splice acceptor site, the "knockout first" cassette must therefore also contain a cryptic splice donor site, which splices to exon 9, restoring expression of all downstream exons. In effect, *Jmjd1c^{k/k}* mice, express normal levels of an intact *Jmjd1c* mRNA. Consequently, blood values of *Jmjd1c^{k/k}* mice were comparable to wt littermate controls (Fig 2D).

To generate a complete ko, we therefore crossed *Jmjd1c^{k/k}* mice with *FLPe* mice, thereby excising the lacZ and neomycin cassettes (*Jmjd1c*-p allele, Fig 1A). The resulting mice were crossed with *Cre deleter* mice, to excise the floxed *Jmjd1c* exons 9 and 10. The resulting transcript contains a premature stop codon within the open reading frame and should therefore be subject to non-sense mediated mRNA decay (*Jmjd1c*-d allele, Fig 1A). However, by RT-PCR we were able to amplify an alternatively spliced, residual mRNA lacking exons 9 and 10 in tissue of *Jmjd1c^{d/d}* mice (Fig 3A). Nonetheless, neither full length nor truncated or deleted Jmjd1c protein was detectable by western blotting in *Jmjd1c^{d/d}* mice (Fig 3B), confirming that the animals are Jmjd1c deficient and residual mRNA was subjected to non-sense mediated decay.

We analyzed peripheral blood counts of *Jmjd1c^{d/d}* mice at 8, 24 and 40 weeks of age (Fig 4A +4B). While there was a trend towards lower platelet values in *Jmjd1c^{d/d}* mice compared to wt controls (Fig 4A), no statistically significant differences were detectable. FACS measurements of the bone marrow (BM) showed no differences in the proportions of the myeloid, erythroid or lymphoid compartment, nor changes in the hematopoietic stem and progenitor populations (Fig 4C–4E). BM cellularity was non-significantly increased in *Jmjd1c^{d/d}* compared to wt littermate controls (S7A Fig). Histopathological analysis of *Jmjd1c^{d/d}* femora revealed a BM composition comparable to wt littermate controls (Fig 4F). Spleen (SPL) weight of *Jmjd1c^{d/d}* mice was significantly reduced after 40 weeks (Fig 4G). Overall survival in *Jmjd1c^{d/d}* mice was similar to wt littermates (Fig 4H). Therefore, absence of *Jmjd1c* either during fetal development or in adult animals had no detectable effect on the hematopoietic system.

Since *Jmjd1c* deficiency did not impact healthy hematopoiesis, we wanted to know whether Jmjd1c is required for Jak2^{V617F}-driven myeloproliferation. *JMJD1C* expression has been reported in hematopoietic cells [10] and we have demonstrated overexpression of *JMJD1C* in patients with MPN. (1) We were able to show *in vitro* that depletion of Jmjd1c by shRNA in *Jak2^{V617F}*-transduced Ba/F3 cells led to a significant decrease of cytokine-independent growth [1]. Given the fact that this model is only partly transferrable because of its artificial nature and the lymphoid background of Ba/F3 cells, we wanted to test the dependency of Jak2^{V617F}-driven MPN on Jmjd1c expression *in vivo*. Hence, we generated animals carrying a Cre-inducible *Jak2^{V617F}* allele either in a wt background or in the context of a *Jmjd1c^{d/d}* knockout. At the age of seven weeks, *Cre* expression was provoked by pI:pC injections, initiating *Jak2^{V617F}* expression. Five weeks following induction of *Jak2^{V617F}* expression blood counts were determined and the animals sacrificed and subjected to histopathological analysis. Complete blood counts of *Jak2^{V617F}* and *Jak2^{V617F} Jmjd1c^{d/d}* animals revealed the same Jak2^{V617F}-driven myeloproliferative phenotype in both genotypes (Fig 5A). No substantial differences in leukocyte or platelet counts were seen between *Jak2^{V617F} Jmjd1c^{d/d}* mice and the *Jak2^{V617F}* control group. Only the red blood cells were significantly increased in *Jak2^{V617F} Jmjd1c^{d/d}* animals compared to their controls (Fig 5A). This finding, however, was not supported by the presence of increased

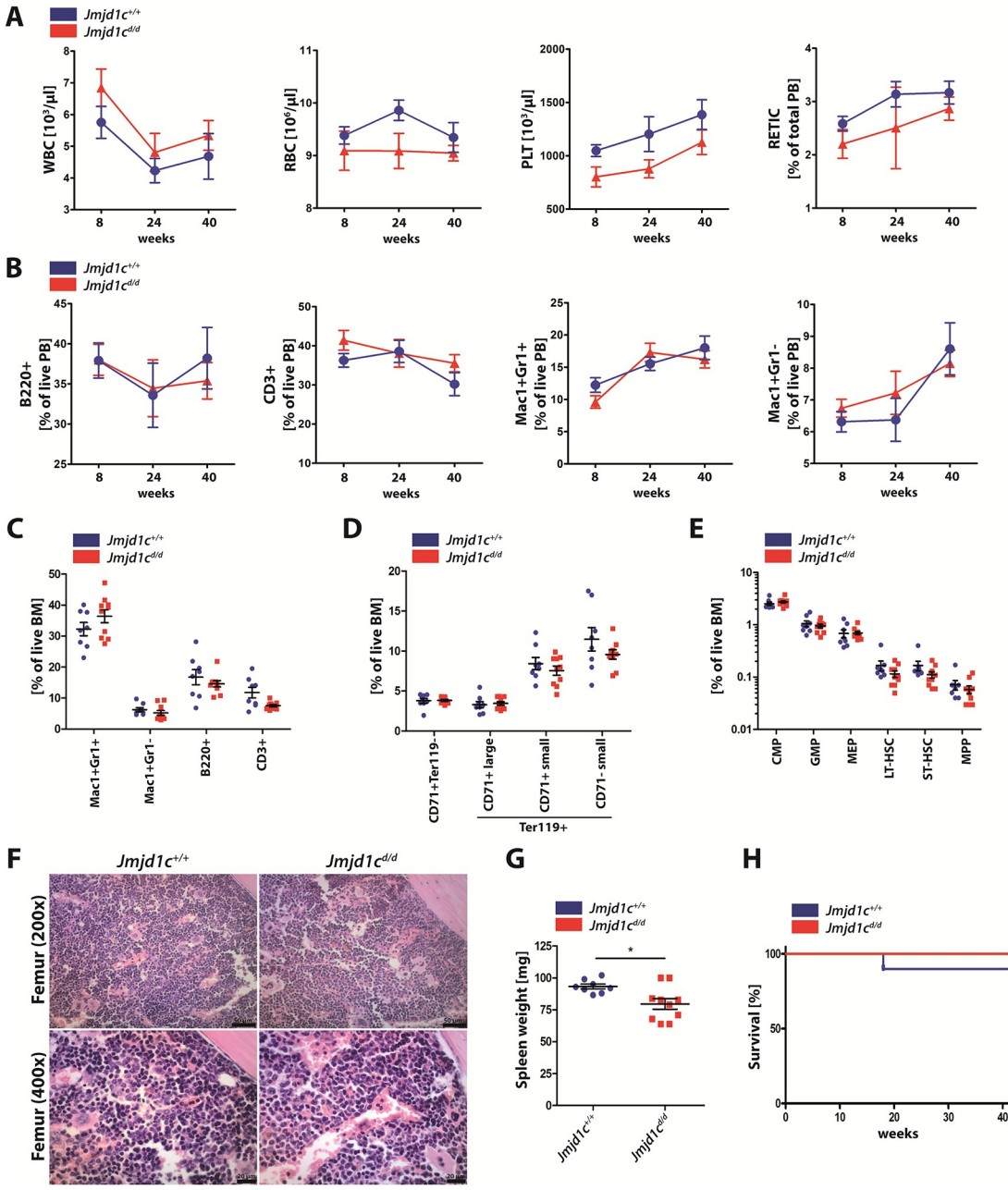

**Fig 4. Loss of Jmjd1c is dispensable for steady-state hematopoiesis. (A-H)** Blood draws of *Jmjd1c$^{d/d}$* mice and wt controls (n = 10 per genotype, lost data points are shown in S2 Table) were performed after 8, 24 and 40 weeks with subsequent sacrifice for final analysis of BM and SPL tissue. **(A)** White blood cell (WBC) count, red blood cell (RBC) count, platelet (PLT) count and the percentage of reticulocytes. **(B)** FACS staining for mature blood cells in PB samples: B cells (B220), T cells (CD3), mature granulocytes (Mac1-Gr1) and monocytes (Mac1). **(C-E)** FACS staining for mature blood cells **(C)**, erythroid precursors **(D)** and HPSC **(E)** in BM samples. **(F)** Representative H&E staining of formalin-fixed and paraffin-embedded femur of wt control (left) and *Jmjd1c$^{d/d}$* mice (right). Scale bars: 50 μm (top) and 20 μm (bottom). **(G)** Spleen weight of 40-week-old mice. **(H)** Survival curve. Data are represented as mean +/- SEM in **A-E** and **G**. *P<0.05. Mann-Whitney U test was used for statistical calculations. Kaplan-Meyer survival analysis was used to determine survival divergences.

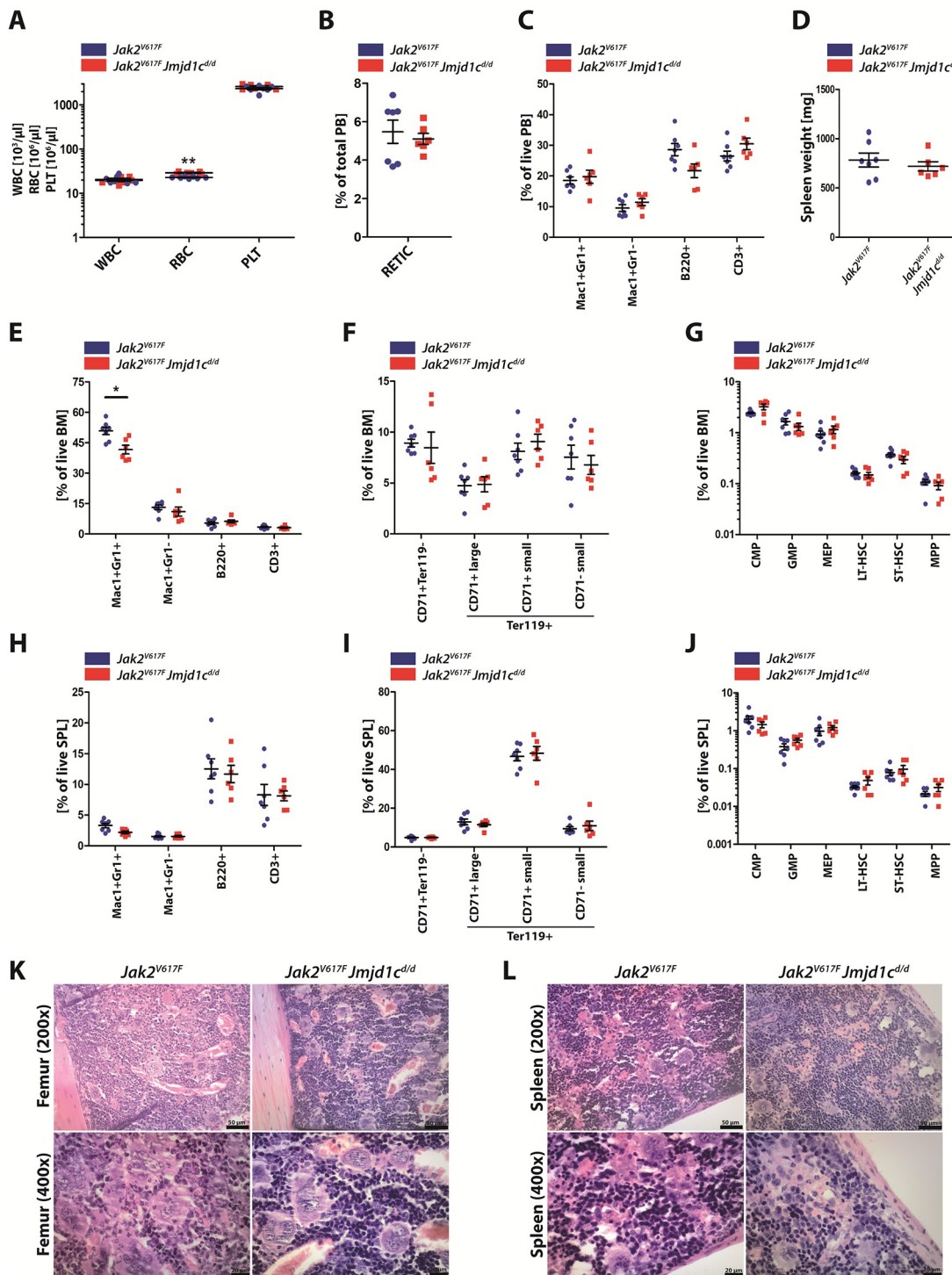

**Fig 5. Loss of Jmjd1c is dispensable for Jak2$^{V617F}$-driven myeloproliferative disease.** (A-L) Analysis of 12-week-old *Jak2$^{V617F}$* and *Jak2$^{V617F}$ Jmjd1c$^{d/d}$* mice (n = 6–7 per genotype, lost data points are shown in S2 Table). (**A**) PB count. (**B**) Reticulocytes in PB samples. (**C**) FACS staining for mature blood cells in PB samples. (**D**) Spleen weight. (**E-G**) FACS staining for mature blood cells (**E**), erythroid precursors (**F**) and HSPC (**G**) in BM samples. (**H-J**) FACS staining for mature blood cells (**H**), erythroid precursors (**I**) and HSPC (**J**) in SPL samples. (**K** and **L**) Representative H&E staining of formalin-fixed and paraffin-embedded femur (**K**) and spleen tissue (**L**) of *Jak2$^{V617F}$* (left) and *Jak2$^{V617F}$ Jmjd1c$^{d/d}$* mice (right). Scale bars: 50 μm (top) and 20 μm (bottom). Data are represented as mean +/- SEM in **A-J**. *P<0.05, **P<0.005. Mann-Whitney U test was used for statistical calculations.

young erythrocytes (reticulocytes, Fig 5B), suggesting that it was not due to an enhanced erythroid drive. Differential analysis of the peripheral blood by FACS measurements revealed that the proportion of mature myeloid and lymphoid cells is unaltered by deletion of *Jmjd1c* (Fig 5C). BM cellularity did not differ between *Jak2*$^{V617F}$ *Jmjd1c*$^{d/d}$ mice and *Jak2*$^{V617F}$ single mutant mice (S7B Fig). Moreover, spleen weight of *Jak2*$^{V617F}$ *Jmjd1c*$^{d/d}$ mice was comparable to that of *Jak2*$^{V617F}$ mice (Fig 5D). Further analysis of the BM (Fig 5E–5G) and the spleen (Fig 5H–5J) revealed no substantial difference between the anticipated *Jak2* mutant phenotype and the *Jmjd1c* ko, with the exception of a reduced number of mature Mac1$^+$ Gr1$^+$ granulocytes in the BM in *Jak2*$^{V617F}$ *Jmjd1c*$^{d/d}$ mice (Fig 5E). Histopathological slides of femora and spleen of *Jak2*$^{V617F}$ *Jmjd1c*$^{d/d}$ animals showed the same myeloproliferation found in femora and spleens of control mice (Fig 5K+5L). These findings support the conclusion that Jmjd1c is dispensable for a Jak2$^{V617F}$-driven myeloproliferative disease.

## Discussion

The role of epigenetic "writers", "readers" and "erasers" in the pathophysiology of human diseases, especially in neoplasias, has gained increased attention in recent years. Because of their enzymatic activity, these proteins constitute potential drug targets. Moreover, contrary to what could have been expected given the ubiquitous importance of epigenetic modifications, the first clinical trials using epigenetic drugs have revealed only moderate clinical side effects [11]. Therefore, understanding the contribution of individual epigenetic enzymes to molecular disease etiology is important to identify novel druggable pathways.

The molecular pathophysiology of myeloproliferative neoplasms is incompletely understood. Most notably, inhibition of JAK2$^{V617F}$, one of three identified driver mutations in MPN patients fails to induce molecular remissions in the majority of patients. We therefore searched for additional drug targets concentrating on epigenetic modifiers as several of the co-occurring mutations in MPN patients affect epigenetic enzymes, for example TET2, DNMT3A and ASXL1.

We and others have shown that the JAK2$^{V617F}$ MPN driver mutation exerts its effect at least in part via up-regulation of the transcription factor NFE2. We have recently shown that among 60 epigenetic modifiers identified as possible downstream NFE2 effectors, the histone demethylase JMJD1C constitutes a novel NFE2 target gene. JMJD1C participates in a positive feedback loop, as it binds the NFE2 locus, thereby increasing NFE2 expression and, in turn, its own transcription. We therefore hypothesized that JMJD1C is required for disease initiation in a Jak2$^{V617F}$-driven murine model of MPN. However, our results clearly demonstrate that in the chosen model, JMJD1C is dispensable for the MPN phenotype.

Two specific limitations of our model must be considered. Firstly, in the constitutive Jmjd1c knock-out strain used, the protein is absent through embryonic development, during which, as we show here (Fig 1B), it is normally robustly expressed. As the JMJD-family of proteins contains 17 highly homologous proteins with demethylase activity, absence of a single member may be compensated by the function of related proteins. In this case, only the deletion of multiple enzymes would visibly alter the Jak2$^{V617F}$-driven murine phenotype.

Secondly, we interrogated whether Jmjd1c is necessary for Jak2$^{V617F}$-driven MPN disease initiation by introducing Jak2$^{V617F}$ into a Jmjd1c-deficient background, perhaps already adapted to this deficiency. However, it is still possible that in an established MPN, *de novo* inhibition of JMJD1C interferes with disease maintenance. Our data showing reduced proliferation of Jak2$^{V617F}$-positive Ba/F3 cells upon JMJD1C inhibition, support this hypothesis. Given the possible redundancy of JMJC domain-containing family proteins, this hypothesis is better addressed by pharmacological inhibition of closely related enzymes than by the genetic

approach targeting a specific protein that we employed. As pan-JMJC domain containing family inhibitors with acceptable toxicity profiles are not yet available, experimental evidence in murine models must await further developments.

We were encouraged to investigate our hypothesis by the published role of JMJD1C in MLL-driven AML stem cells. However, our data may also reflect a fundamental biological difference between murine models of MLL-driven AML and JAK2$^{V617F}$-driven MPN. JMJD1C and KDM3B inhibitors exert their effect mainly through induction of differentiation [12]. By nature of the diseases modeled, differentiation is heavily disturbed in MLL rearranged leukemias but it is not affected in MPN. Similarly, the therapeutic use of retinoids in APL, induction of differentiation by JMJD-inhibition may therefore hold therapeutic potential in AML but not in MPN.

There is growing evidence that JMJD1C plays an important role in myogenesis ([13] and Fig 1), highlighted by the fact that expression of MyoD, a key regulator of myogenesis, is dependent on Jmjd1c. The demethylase decreases H3K9me$^2$ marks at the MyoD locus, thereby increasing its transcription [13]. Similarly, JMJD1C is a target of DPF3b, a transcriptional activator important for heart and muscle development, that associates with the BAF chromatin remodeling complex [14]. Our observation of strong embryonic Jmjd1c expression in developing muscle tissues suggests additional roles for JMJD1C in organs besides the hematopoietic system.

## Supporting information

**S1 Fig. Transcript variants for the *Jmjd1c* gene.** Exon numeration for protein coding variant 1 (Ensembl: ENSMUST00000174408.7, NCBI: NM_207221.2) and protein coding variant 2 (Ensembl: ENSMUST00000173689.7, NCBI: NM_001242396.1). This article uses the numeration of variant 1. Other articles might use variant 2 for exon numeration [3]. Exons removed in the d allele of *Jmjd1c* knockout mice are highlighted in red. This figure is based on the Ensembl genome browser.
(PDF)

**S2 Fig. Genotyping of *Jmjd1c$^{k/k}$* mice. (A)** Schematics of the knockout first approach (*Jmjd1c*-k allele). **(B)** Genotyping PCR with genomic DNA extracted from ear tissue.
(PDF)

**S3 Fig. Genotyping PCRs of *Jmjd1c$^{d/d}$* mice. (A)** Primer locations for genotyping PCRs. **(B)** Multiplex PCR for genotyping using material obtained from bone marrow (BM), testicular tissue (TE) or peripheral blood (PB).
(PDF)

**S4 Fig. Original, minimally cropped and adjusted gel and blot images. (A-C)** Full gel of picture shown in Fig 2C **(A)**, Fig 3A **(B+C)**. **(B)** Depicts wild type splicing. **(C)** Gel showing alternate splicing. **(D+E)** Uncropped wb presented in Fig 3B probed with an antibody against Jmjd1c **(D)** or β-actin **(E)**. Lanes labeled with "unrelated experiments" have been previously published by us [1]. **(F+G)** Gel pictures relating to S2 Fig **(F)** and S3 Fig **(G)**.
(PDF)

**S5 Fig. DNA sequence of the 115 bp (SA) insert.** This supplemental figure relates to Fig 2B +2C. The last 25 bp of exon 8 and the first 25 bp of exon 9 are highlighted in grey. The 115 bp (SA) insert is highlighted in orange.
(PDF)

**S6 Fig. Expression sites of *Jmjd1c* during embryonic development.** Sections showing beta-galactosidase signals during kidney development **(A)**, neurogenesis **(B)**, lung development

(C) and development of the intestinal tract (D). (A-D) E11.5 and E13.5 whole mouse embryos were beta-galactosidase stained with subsequent formalin-fixation, paraffin embedding and sectioning, while E17.5 were first frozen followed by beta-galactosidase staining of the cryosections. All sections were counterstained with Nuclear Fast Red. (E) Background beta-galactosidase activity in the intestine of E17.5 embryos.
(PDF)

**S7 Fig. Bone marrow cellularity. (A)** Bone marrow cellularity in *Jmjd1c*$^{+/+}$ and *Jmjd1c*$^{d/d}$ mice (n = 10 per genotype, lost data points are shown in S2 Table) after 40 weeks. The cell count of one femur was added to the cell count of one tibia. **(B)** Bone marrow cellularity in *Jak2*$^{V617F}$ and *Jak2*$^{V617F}$ *Jmjd1c*$^{d/d}$ mice (n = 6–7 per genotype) after 12 weeks. The cell count of one femur was added to the cell count of two tibiae. **(A)** and **(B)** Mann-Whitney U test was used for statistical calculations.
(PDF)

**S1 Table. Primer sequences.**
(DOCX)

**S2 Table. Lost data points.**
(DOCX)

# Acknowledgments

The authors sincerely thank Franziska Zipfel for expert technical assistance. H.F.S. likes to express his gratitude to the M.D. mentoring program MOTI-VATE.

# Author Contributions

**Conceptualization:** Hans F. Staehle, Albert Gruender, Heike Luise Pahl, Jonas Samuel Jutzi.

**Data curation:** Hans F. Staehle, Johannes Heinemann, Jonas Samuel Jutzi.

**Formal analysis:** Hans F. Staehle, Johannes Heinemann, Anne M. Omlor, Jonas Samuel Jutzi.

**Funding acquisition:** Heike Luise Pahl, Jonas Samuel Jutzi.

**Investigation:** Hans F. Staehle, Johannes Heinemann, Anne M. Omlor, Jonas Samuel Jutzi.

**Methodology:** Hans F. Staehle, Johannes Heinemann, Albert Gruender, Anne M. Omlor, Heike Luise Pahl, Jonas Samuel Jutzi.

**Project administration:** Hans F. Staehle, Albert Gruender, Heike Luise Pahl, Jonas Samuel Jutzi.

**Resources:** Heike Luise Pahl, Jonas Samuel Jutzi.

**Software:** Hans F. Staehle, Albert Gruender, Heike Luise Pahl, Jonas Samuel Jutzi.

**Supervision:** Albert Gruender, Heike Luise Pahl, Jonas Samuel Jutzi.

**Validation:** Albert Gruender, Heike Luise Pahl, Jonas Samuel Jutzi.

**Visualization:** Hans F. Staehle, Johannes Heinemann, Anne M. Omlor.

**Writing – original draft:** Hans F. Staehle, Heike Luise Pahl, Jonas Samuel Jutzi.

**Writing – review & editing:** Albert Gruender, Heike Luise Pahl, Jonas Samuel Jutzi.

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
