## [Decision Letter · Decision Letter 0]

3 Dec 2019

PONE-D-19-29589

Jmjd1c is dispensable for healthy adult hematopoiesis and Jak2V617F-driven myeloproliferative disease initiation in mice

PLOS ONE

Dear Dr. Jutzi,

Thank you for submitting your manuscript to PLOS ONE. After careful consideration, we feel that it has merit but does not fully meet PLOS ONE’s publication criteria as it currently stands. Therefore, we invite you to submit a revised version of the manuscript that addresses the points raised during the review process.

Please add any additional details regarding the hematopoietic compartment of these knockout mice as suggested by the reviewer.

We would appreciate receiving your revised manuscript by Jan 17 2020 11:59PM. To enhance the reproducibility of your results, we recommend that if applicable you deposit your laboratory protocols in protocols.io, where a protocol can be assigned its own identifier (DOI) such that it can be cited independently in the future. For instructions see: http://journals.plos.org/plosone/s/submission-guidelines#loc-laboratory-protocols

We look forward to receiving your revised manuscript.

Kind regards,

Kevin D Bunting

Academic Editor

PLOS ONE

Journal Requirements:

2. To comply with PLOS ONE submission guidelines, in your Methods section, please provide additional information regarding your statistical analyses. For more information on PLOS ONE's expectations for statistical reporting, please see https://journals.plos.org/plosone/s/submission-guidelines.#loc-statistical-reporting.

Reviewers' comments:

Reviewer's Responses to Questions

**Comments to the Author**

1. Is the manuscript technically sound, and do the data support the conclusions?

Reviewer #1: Yes

2. Has the statistical analysis been performed appropriately and rigorously? 

Reviewer #1: Yes

3. Have the authors made all data underlying the findings in their manuscript fully available?

Reviewer #1: Yes

4. Is the manuscript presented in an intelligible fashion and written in standard English?

Reviewer #1: Yes

5. Review Comments to the Author

Reviewer #1: In this manuscript, Staehle et al. investigated JMJD1C’s role in steady state hematopoiesis and JAK2V617F driven MPN. This study stems from their previous finding that JMJD1C is a target of NFE2, a transcription factor important for MPN, and acts in positive feedback loop to promote NFE overexpression in MPNs. The authors did a thorough validation of genetic Jmjd1c mouse model used. Their study showed that JMJD1C is dispensable for steady state hematopoiesis using a whole body knockout, largely similar to results from a previous published one. They further characterized the expression pattern of JMJD1C during embryogenesis. Moreover, they showed that JMJD1C loss is dispensable for JAK2V617F driven MPN. This study collaborated previous published hematopoietic phenotype of Jmjd1c knockout using a different Cre, expanded our understanding of the role of JMJD1C during early development and its role in MPN development. Overall, the results are supported by experiments that are well designed and performed with rigorous statistics.

Major points

1. Have the authors examined whether there is any bone marrow cellularity changes in Jmjd1cd/d mice compare to Jmjd1c+/+ mice as shown by the previous publication using Vav1Cre? If so, are there any changes in absolute cell numbers of mature and HSPC compartments in the bone marrow?

2. As the authors pointed out in the Discussion section, it is possible that adaptation to JMJD1C loss may explain the lack of phenotype upon JAK2V617F induction in JMJD1C null background. It still remains possible that JMJD1C is required for the maintenance but not initiation of MPN. Along this line, no effect was observed on MLL-AF9 leukemia in a Jmjd1cf/f Vav1Cre background in the previous study. Have the authors tried to use conditional rather than null allele of Jmjd1c to address this possibility?

3. Both Jak2V617F and Jmjd1c deletion by themselves have been shown to reduce BM cellularity. Are there any changes in bone marrow cellularity in Jak2V617FJmjd1cd/d compare to Jak2V617F mice? If so, what are the changes in absolute cell numbers of mature and HSPC compartments?

Minor points

1. Line 92-93, “shRNA-mediated JMJD1C depletion..”: reference cited used genetic model not shRNA.

6. PLOS authors have the option to publish the peer review history of their article (what does this mean?). If published, this will include your full peer review and any attached files.

Reviewer #1: No

---

## [Author Response · Author response to Decision Letter 0]

10 Jan 2020

To

Prof. Dr. Kevin Bunting 

PLOS ONE

Boston, January 9th, 2020

Dear Prof. Bunting, dear Kevin, 

Thank you for reviewing our manuscript entitled “Jmjd1c is dispensable for healthy adult hematopoiesis and Jak2V617F-driven myeloproliferative disease initiation in mice”.

We addressed the reviewer’s comments point-by-point, see responses below. Moreover, in line with PLOS ONE’s journal requirements and guidelines, we included some additional information, including all original gel pictures into the revised manuscript.

Reviewer’s comments: In this manuscript, Staehle et al. investigated JMJD1C’s role in steady state hematopoiesis and JAK2V617F driven MPN. This study stems from their previous finding that JMJD1C is a target of NFE2, a transcription factor important for MPN, and acts in positive feedback loop to promote NFE overexpression in MPNs. The authors did a thorough validation of genetic Jmjd1c mouse model used. Their study showed that JMJD1C is dispensable for steady state hematopoiesis using a whole body knockout, largely similar to results from a previous published one. They further characterized the expression pattern of JMJD1C during embryogenesis. Moreover, they showed that JMJD1C loss is dispensable for JAK2V617F driven MPN. This study collaborated previous published hematopoietic phenotype of Jmjd1c knockout using a different Cre, expanded our understanding of the role of JMJD1C during early development and its role in MPN development. Overall, the results are supported by experiments that are well designed and performed with rigorous statistics.

Major points:

1. Have the authors examined whether there is any bone marrow cellularity changes in Jmjd1cd/d mice compare to Jmjd1c+/+ mice as shown by the previous publication using Vav1Cre? If so, are there any changes in absolute cell numbers of mature and HSPC compartments in the bone marrow?

Response to 1.:

Indeed, we examined the total bone marrow (BM) cellularity in the Jmjd1c d/d mice and compared it to Jmjd1c +/+ mice. We now include these data as supplemental figure 7A in the revised manuscript. There was no statistically significant difference in BM cellularity. We did not examine cellularity of sub-compartments of HSPCs.

2. As the authors pointed out in the Discussion section, it is possible that adaptation to JMJD1C loss may explain the lack of phenotype upon JAK2V617F induction in JMJD1C null background. It still remains possible that JMJD1C is required for the maintenance but not initiation of MPN. Along this line, no effect was observed on MLL-AF9 leukemia in a Jmjd1cf/f Vav1Cre background in the previous study. Have the authors tried to use conditional rather than null allele of Jmjd1c to address this possibility?

Response to 2.:

Thank you for this question. Indeed, while we examined the diseases initiation process by Jak2V617F induction in the Jmjd1c d/d background, we have not knocked out Jmjd1c in an already established Jak2V617F-driven disease background. We established a constitutive Jak2V617F model which we planned to use for crossing with the inducible Jmjd1c ko mouse model. While we obtained viable Jak2V617F offspring, it was almost impossible to maintain the line itself due to fertility problems of the females. Crossing them to Jmjd1c fl/fl mice proved unfeasible. It is known in literature and we have shown as well (Jutzi et al., HemaSphere, 2018) that floxed Jak2V617F mice in an Mx1Cre background already show a full-blown phenotype without previous induction, probably due to a pro-inflammatory feed-forward activation of the Mx promoter, followed by cre recombinase expression. This knowledge thwarted our alternative approach of using Jmjd1 fl/fl BM in a retroviral Jak2V617F model followed by pi:pC induction as it would not be a clean model to study disease maintenance. The leakiness would cause Jmjd1c excision by the cre recombinase already prior to retroviral infection with the Jak2 mutation. We therefore focused on disease initiation entirely. 

3. Both Jak2V617F and Jmjd1c deletion by themselves have been shown to reduce BM cellularity. Are there any changes in bone marrow cellularity in Jak2V617FJmjd1cd/d compare to Jak2V617F mice? If so, what are the changes in absolute cell numbers of mature and HSPC compartments?

Response to 3.:

We have determined the total BM cellularity of both genotypes and have not found a statistically significant difference between Jak2V617F and Jak2 V617F Jmjd1c d/d mice. We have not further subdivided the examined cellularity into the HSPC sub-compartments. We included the data in the revised manuscript as supplemental figure 7B.

Minor points:

1. Line 92-93, “shRNA-mediated JMJD1C depletion...”: reference cited used genetic model not shRNA.

Thank you for pointing out this error. Indeed, it is of course a genetic deletion of Jmjd1c with a comparable approach as ours. We have corrected the paragraph in the introduction in the revised manuscript and apologize for the mistake made.

Journal Requirements:

Response to 1.: We have now addressed all the PLOS ONE style requirements, in the cases where we hadn’t addressed it before (e.g. title page).

2. To comply with PLOS ONE submission guidelines, in your Methods section, please provide additional information regarding your statistical analyses. For more information on PLOS ONE's expectations for statistical reporting, please see https://journals.plos.org/plosone/s/submission-guidelines.#loc-statistical-reporting.

Response to 2.: In line with PLOS ONE guidelines, we expanded the statistical section in the methods part of the manuscript. 

Response to 3.: We now included the data on beta-galactosidase staining previously referred to as “data not shown” in the revised manuscript: We included a “negative control” for the staining in supplemental figure 6E to show that there is background activity in the intestine at time point E17.5. 

Missing data points: The whole study took us more than 12 months after establishing the mouse lines. This is due to the size of the cohort and expected Mendelian ratios as well as the long follow-up of up to 40 weeks. In light of that, we were not able to obtain all data points from all mice at all time points, mostly due to technical errors or premature deaths of single mice. In the revised manuscript, we now include a supplemental table 2 that lists all lost data points and the specific reasons for it. 

4. PLOS ONE now requires that authors provide the original uncropped and unadjusted images underlying all blot or gel results reported in a submission’s figures or Supporting Information files. This policy and the journal’s other requirements for blot/gel reporting and figure preparation are described in detail at https://journals.plos.org/plosone/s/figures#loc-blot-and-gel-reporting-requirements and https://journals.plos.org/plosone/s/figures#loc-preparing-figures-from-image-files. When you submit your revised manuscript, please ensure that your figures adhere fully to these guidelines and provide the original underlying images for all blot or gel data reported in your submission. See the following link for instructions on providing the original image data: https://journals.plos.org/plosone/s/figures#loc-original-images-for-blots-and-gels. In your cover letter, please note whether your blot/gel image data are in Supporting Information or posted at a public data repository, provide the repository URL if relevant, and provide specific details as to which raw blot/gel images, if any, are not available. Email us at plosone@plos.org if you have any questions.

Response to 4.: Following the guidelines, we included original gel pictures for all gels included in the main figures, into a new supplemental figure 4 (S4 Fig.). Because of this, there is a shift in the numbering of the originally submitted supplemental figures. All changes are highlighted in red. All figures meet the journal’s requirement now, including visualization of individual data points (figure 4 C-E, G and figure 5A-J).

Important to note: The western blot membrane used for experiments depicted in figure 3B and revised supplemental figure 4 D+E has been previously used for unrelated experiments, in which it was also probed with the JMJD1C antibody. The lanes labeled “unrelated experiments” show JMJD1C and actin bands of samples used for a different publication from our lab: Peeken et al., Blood, 2018.

Additional changes and remarks by the authors:

Copy and paste error in figure 5I:

During the process of harmonizing our data to the journal’s requirement, meaning showing individual data points, we realized an error in the uploaded figure 5. While the content of the results section and the figure legend was correct, a copy and paste error occurred and figure 5H was shown twice. We have now replaced figure 5I with the originally intended data showing erythropoietic cells defined by flow in the spleen of Jak2V617F and Jak2V617F Jmjd1cd/d mice. We would like to apologize for this mistake!

Spleen weight of Jmjd1c d/d:

Using the Mann-Whitney U test, the spleen weights of Jmjd1c d/d are indeed statistically significantly lower when compared to WT littermate controls. However, we failed to mark the statistical significance in the initially submitted manuscript. We sincerely apologize for this mistake. The revised figure 4G contains the asterisk to indicate a p value of <.05. We inserted a corrected statement into the results section, highlighted in red. However, this does not change the conclusion that we find only minor differences with questionable biological implications.

We hope that with these changes made and additional data included, you consider our manuscript worthy of publication in PLOS ONE.

Thank you for your time and effort in reviewing the manuscript.

Sincerely, 

 Jonas S. Jutzi, MD, PhD

---

## [Editor Report · Decision Letter 1]

14 Jan 2020

Jmjd1c is dispensable for healthy adult hematopoiesis and Jak2V617F-driven myeloproliferative disease initiation in mice

PONE-D-19-29589R1

Dear Dr. Jutzi,

We are pleased to inform you that your manuscript has been judged scientifically suitable for publication and will be formally accepted for publication once it complies with all outstanding technical requirements.

With kind regards,

Kevin D Bunting

Academic Editor

PLOS ONE
---

## [Editor Report · Acceptance letter]

15 Jan 2020

PONE-D-19-29589R1 

Jmjd1c is dispensable for healthy adult hematopoiesis and Jak2^V617F^-driven myeloproliferative disease initiation in mice 

Dear Dr. Jutzi:

I am pleased to inform you that your manuscript has been deemed suitable for publication in PLOS ONE. Congratulations! Your manuscript is now with our production department. 

With kind regards,

on behalf of

Dr. Kevin D Bunting 

Academic Editor

PLOS ONE